# What enabled the successful implementation of a quality certification initiative in Bhavnagar, Gujarat? A policy analysis case study

**K. Shruti Lekha**[1], **Sudha Ramani**[2], **Harsha Joshi**[2*], **Preet Verma**[3], **Sumit Malhotra**[3], **Tapasvi Puwar**[1], **Chandramani Kumar**[4], **Ankita Shah**[5], **Anish Sinha**[1], **Deepak Saxena**[1]

**1** Indian Institute of Public Health, Gandhinagar, Gujarat, India, **2** Johns Hopkins India Private Limited, New Delhi, India, **3** Centre for Community Medicine, All India Institute of Medical Sciences, New Delhi, India, **4** District Health Office, Bhavnagar Jilla Panchayat, Bhavnagar, Gujarat, India, **5** State Health Systems Resource Centre, Gandhinagar, Gujarat, India

* hjoshi2@jh.edu

## Abstract

High-quality health systems are key to improving population health outcomes globally. In India, the National Quality Assurance Standards (NQAS) is a certification policy adopted by the government to improve the quality of care in public health facilities. This policy aims to assess public health facilities through a set of comprehensive, pre-defined standards derived from global best practices. However, only a small number of districts in the country have been able to effectively complete certifications as mandated. Bhavnagar, a district in the state of Gujarat in western India, is a positive deviant that has certified the majority of its primary health facilities. This study attempts to delineate factors that have led to successful quality certifications in Bhavnagar. Qualitative data was collected between December 2023–February 2024, and includes in-depth interviews of staff from state, district, and facility levels (n = 20), and group discussions with facility staff (n = 2). Data has been analysed from the lens of the 'policy triangle', comprising actors (policy-makers, managers, implementers), context (political support), content (the policy and interpretation), and processes (plans, implementation, and evaluation). We found that Bhavnagar's political context is supportive of quality certifications, with the district's top managers directing the certification process. The district's mid-level operational team on quality has engaged with innovative solutions to solve checklist-related hurdles in infrastructure like establishing a temporary fire-escape or installing screens between rooms for additional space. A peer-mentoring system, wherein staff from already certified primary health facilities act as mentors to prospective ones, has been instituted. This study consolidates empirical lessons for boosting quality certifications in similar contexts. Further, it engages with quality as not just a technical issue, but a political one that is dependent on actors, their relationships, and the implementation context. In doing so, it deepens current understandings of quality improvement strategies in health systems globally.

**Data availability statement:** The raw data for this study are transcripts of interviews, which contain potentially identifying and sensitive information about facilities and staff. We are unable to share these data publicly because of restrictions by the Institutional Review Boards of the two institutes involved, as participants did not consent to sharing of their data outside of the study team. Relevant, de-identified excerpts of the transcripts are included in the paper. Interested readers can request for data to the Institutional Ethics Committee at Indian Institute of Public Health, Gandhinagar (Email: iiphgiec@iiphg.org).

**Funding:** This study was supported by Bill and Melinda Gates Foundation (https://www.gatesfoundation.org/) grant to the International Health Department at Bloomberg School of Public Health, Johns Hopkins University for India Primary Healthcare Support Initiative project (Grant Number: 135043). Indian Institute of Public Health, Gandhinagar is a sub-grantee of this project with DS as the principal investigator at the institute. The funders had no role in study design, data collection and analysis, decision to publish, or preparation of the manuscript.

**Competing interests:** The authors have declared that no competing interests exist.

## Introduction

Gaps in the quality of healthcare pose a threat to the achievement of universal health coverage and remains an issue of public health concern across the globe [1,2]. Merely focusing on the expansion of service coverage in health systems without improvements in quality is 'ineffective, wasteful, and unethical' [3]. Recognising this, ministries of health across countries have adopted a wide range of quality enhancement strategies in health systems. These span improvements at the macro-levels of health systems (governance and system-wide) as well as micro-levels (training of providers, improvements in service delivery, and in facilities available to the community) [3]. In addition, various strategies for the standardised assessments of quality in healthcare such as licensing, certification, and accreditation have also been widely attempted by health ministries [4].

Aligned with global thinking, the Ministry of Health and Family Welfare (MoHFW), government of India, has also instituted numerous certification initiatives to improve the quality of public health services. Among these, three quality certification initiatives have gained political prominence in the recent years- the National Quality Assurance Standards (NQAS) (2013) [5], the Kayakalp scheme (2015) [6], and the Labour Room Quality Improvement Initiative (LaQshya) (2017) [7]. These certification initiatives aim at improving service delivery in public health facilities through strengthening existing structures and operations. While all three initiatives are centrally driven by the MoHFW, they are implemented by different state governments to varying degrees [8].

Despite the enthusiastic adoption of the aforementioned quality certification initiatives in India by the MoFHW, data suggests that their local implementation has been sluggish. For instance, only 6.4% of the total rural primary health facilities (1596 of 24918) in India have been certified through the NQAS initiative [9]. Anecdotal reports and state-level discussions on implementation barriers to certification note issues with the utilization of funding available for certifications, manpower shortages in health facilities that do not meet the standards demanded by such initiatives, and competing priorities of staff that take focus away from these initiatives [10–12]. Notwithstanding these barriers, some states (and districts within states) have done better than others in terms of obtaining quality certifications. For example, the states of Andhra Pradesh and Gujarat have achieved higher numbers of NQAS certifications, and the states of Madhya Pradesh and Maharashtra have achieved higher numbers of LaQshya certifications, in comparison to other states in the country [9].

This study is centred around a 'positively deviant' district in Gujarat, Bhavnagar (see Fig 1), that has made exceptional strides in quality improvement. Bhavnagar has completed the process of NQAS certification for three-quarters of its rural primary health care facilities (tier-1 peripheral health facilities that cater to 30,000 population [13]). Fig 2 depicts the progress of the district in NQAS certifications from September 2020 onwards. More than 80% of the district's rural primary health care facilities have been NQAS certified. These numbers are one of the highest numbers of certifications that any district in India has achieved in a short timeframe. Positive deviance approaches are often used in the development sector to assess programs critically and understand factors underlying the success of interventions. Such approaches, in addition to the understanding of barriers, enable the design of plans and actions and allow for the replication of successful interventions in other contexts [14,15].

In this study, we explore the range of factors that have led to the successful certification of majority of primary health facilities in Bhavnagar. Further, we examine how the NQAS certification process in Bhavnagar has contributed to improving the quality of care at primary health care facilities, as perceived by key informants in the district.

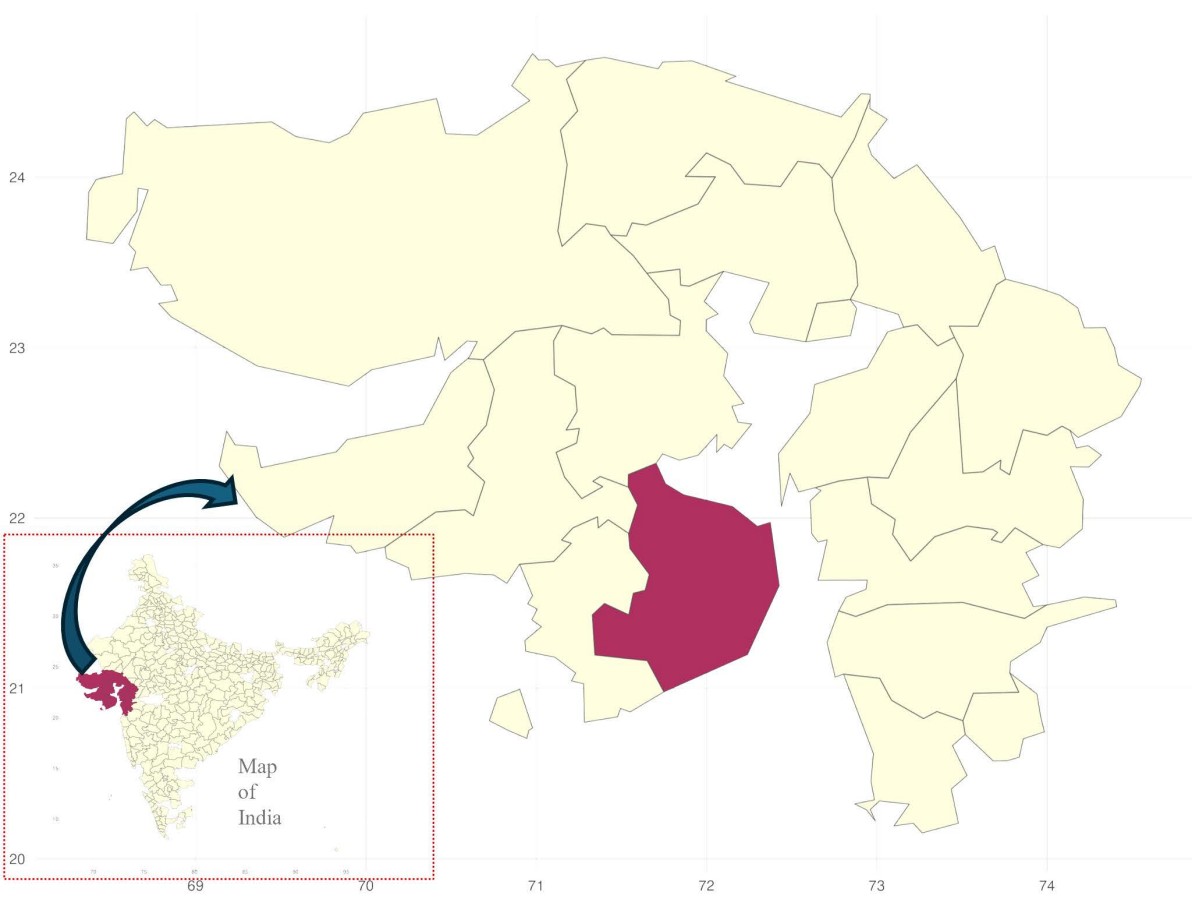

**Fig 1. Map of Gujarat state highlighting location of the Bhavnagar district.** (Source of the basemap shapefile: https://www.indianremo-tesensing.com/2017/01/Download-India-shapefile-with-kashmir.html).

This study is important in two ways. One, empirical lessons from Bhavnagar can help to consolidate ideas that have the potential to boost certification processes in other districts in the country. Currently, there is less understanding of why some states (and some districts within states) have made more progress than others in obtaining certifications. While there has been some documentation of the performance of such certification initiatives in India, particularly in the form of government reports [9], less academic attention has been paid to the factors that affect the implementation of these strategies on the ground. Reports have discussed barriers to certification such as the lack of teamwork, competing priorities of staff, and issues with funding [10,11]. A recent case study of the quality initiative, LaQshya, suggests the need for strong leadership, training in quality-related competencies, and improvement in data management to improve its implementation [12]. However, to the best of our knowledge, empirical data on the range of factors that can enable successful quality certification is currently lacking from India. Second, it is increasingly being recognised that improvement of quality is not just a technical issue, but a political one [16]. Despite this recognition, there is less literature that engages with quality as a political issue, that is dependent on actors and the way they frame and communicate ideas, relationship between actors, as well as the implementation context. Successful policies can be considered as ones that ultimately manage to gain a place in local action agendas [17]. In this study, we use one of the seminal policy analysis

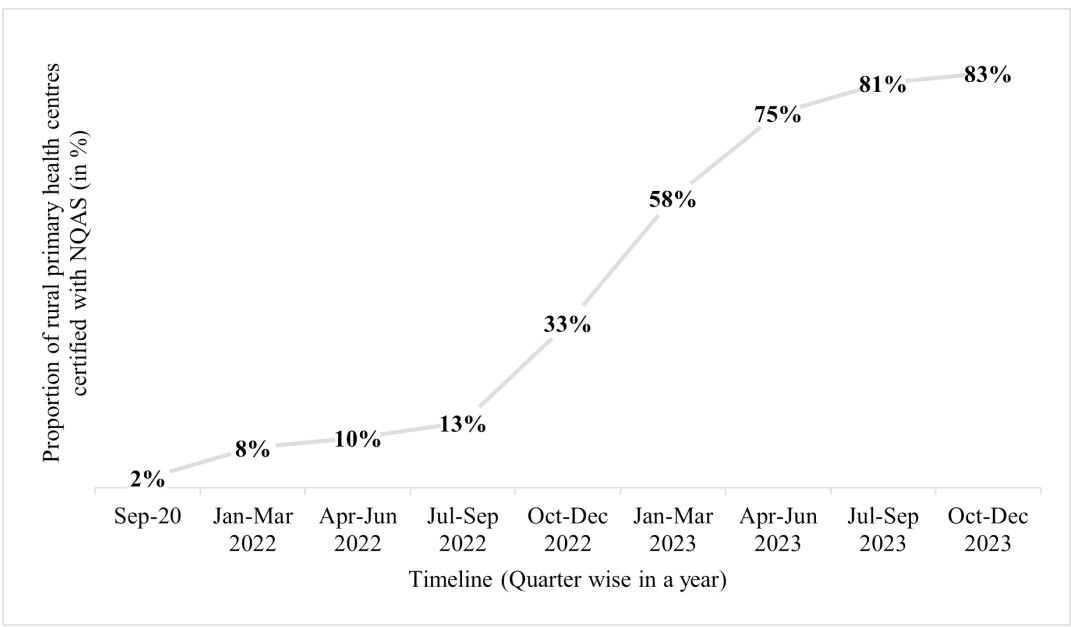

**Fig 2. Progress of NQAS certification of primary health facilities in Bhavnagar district.** (Source: Data from the district administration in Bhavnagar).

frameworks, the policy triangle [18] to deepen our understandings of quality improvement in health systems as a political process.

## Background: The public health system in India and NQAS certification

India has a tiered public health system, comprising primary, secondary and tertiary care facilities [19]. Primary health care facilities are the entry point into this system, and provide integrated preventive, curative and promotive care to the community. Each primary health care facility is to be manned by one medical doctor, staff nurses, and community health workers [13]. Secondary level care is provided by district-level hospitals and tertiary care by specialty hospitals [19].

The operational guidelines of the MoHFW, India on quality [8] has adopted the Institute of Medicine's definition of quality as "*the degree to which health care services for individuals and populations increase the likelihood of desired health outcomes and are consistent with current professional knowledge*" [20]. The NQAS certification initiative of the MoFHW is aimed at facilities at both primary and secondary levels in the public health system [8]. This study's focus is on NQAS certifications for primary health care facilities only. Assessments are done through a set of comprehensive, pre-defined standards. These standards incorporate global best practices across various quality frameworks but have been adapted to suit the public health system in the country. Public health facilities can use these standards for self-assessment, and for obtaining state as well as national level certification through a set application and assessment system. The assessments are carried out using a standardized checklist, arranged under eight broad themes- service provision, patient rights, inputs, support services, clinical services, infection control, quality management and system-level outcomes. The assessments result in a quality score being given to a facility, and facilities that have at least 70% scores get certified. NQAS certified facilities get both financial and non-financial incentives linked to the quality scores they receive during assessment. A facility gets

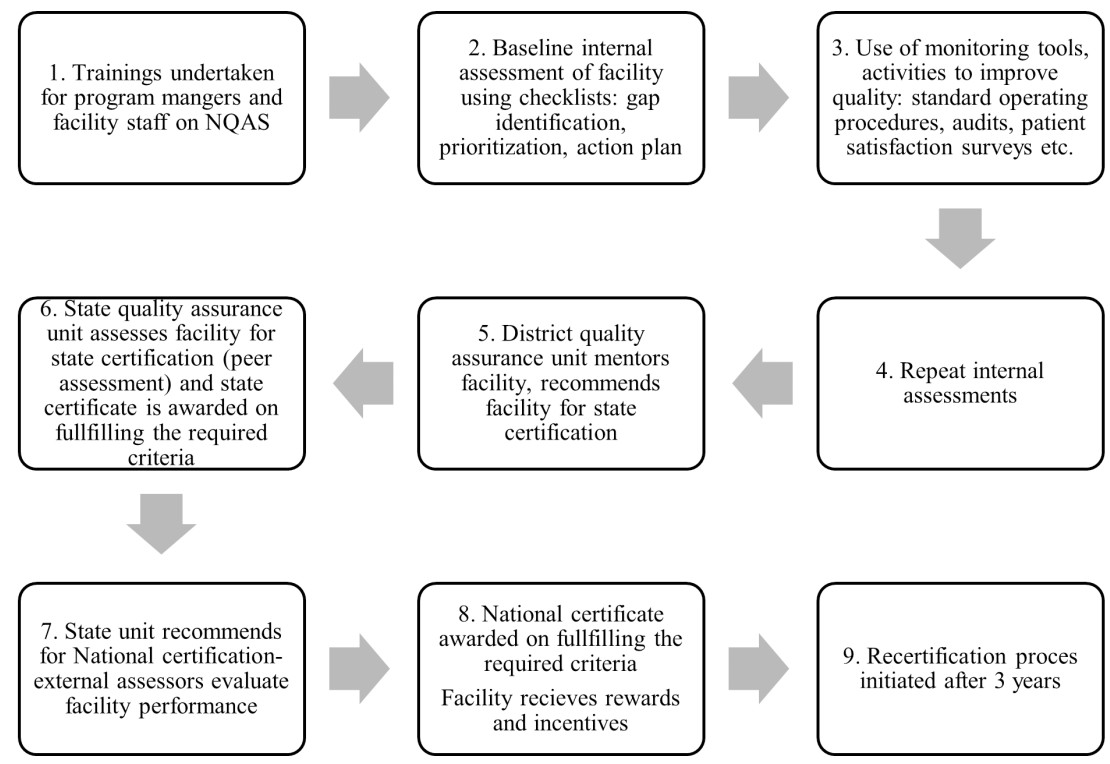

**Fig 3. NQAS certification process (adapted from the operational guidelines on NQAS, MoHFW) [8].**

certified for three years after which the facility needs to re-apply for NQAS certification. Fig 3 depicts details of the certification process.

## Methods

Bhavnagar is a coastal district of Gujarat with a total population of 2.4 million [21]. There are 10 blocks (Talukas) with 48 tier-1 primary health care facilities (that cater to 30000 population) and 300 sub-centers under them (that cater to 5000 population) (Table 1) [22]. These facilities are managed by the Chief District Health Officer office under the Zilla panchayat (district council).

**Table 1. Bhavnagar district profile [21,22].**

| Population characteristics | |
|---|---|
| Total population | 2.4 million |
| Urban population (%) | 41 |
| Rural population (%) | 59 |
| **Health infrastructure** | |
| Medical college hospital | 1 |
| Referral centers | 18 |
| Primary health care facilities (rural) | 48 |
| Primary health care facilities (urban) | 14 |
| Sub-health centres | 300 |

This is a cross-sectional qualitative study, in which nineteen in-depth interviews and two focus group discussions was carried out with staff in the public health system in Bhavnagar. Data collection for the study was done between December 2023 and February 2024. Details of the methodology followed are below:

## Study sample and data collection

In depth interviews were conducted with the officials from the state quality assurance cell, the district health team, and staff from primary health care facilities that have obtained the quality certification (see Table 2 for topics and Tables 3 and 4 for participant details). To supplement the interviews, two Focus Group Discussions (FGDs) were also carried out with staff from primary health care facilities that have completed certification for NQAS. The tools used for the study are attached (S1 File).

At the state and district level, we purposively sampled staff engaged closely with quality certification processes. One of the district level interviews was with a 'policy entrepreneur' [23], who was considered indispensable in scaling up NQAS processes in Bhavnagar. During these interviews, we encouraged staff to speak about their experiences with NQAS and sought their opinions on what was done differently in Bhavnagar to overcome usual barriers to certification.

**Table 2. Topics covered in data collection.**

| Policy-level interviews | Staff interviews at primary healthcare facilities |
|---|---|
| • Selection of the facilities for certification<br>• Planning and funding for certifications<br>• Training and mentoring plans for facilities<br>• Opinions on what factors contributes to successful certifications.<br>• Perceived advantages and disadvantages of certification<br>• Perceptions of sustainability of quality improvements | • Roles in the certification process<br>• Using the checklist- lessons learnt and practical adaptation<br>• Experience of training and peer-mentoring<br>• Experiences of the process of certification<br>• Perceived advantages and disadvantages of certification<br>• Perceptions of sustainability of quality improvements |

**Table 3. Data collection: Participant details, program managers.**

| Sr. No. | Participants | Participant number (In-depth interview) |
|---|---|---|
| 1 | State- level manager | 01 |
| 2 | District and sub-district level managers | 08 |

**Table 4. Data collection: Participant details, staff at primary healthcare facilities (in brackets- participant number).**

| Facility number | In-depth interview | Focus group discussions |
|---|---|---|
| Facility 1 | Doctor (1) | Support staff[a] (5 members) |
| Facility 2 | Doctor (1) | – |
| | Support staff (3) | – |
| Facility 3 | Doctor (1) | – |
| | Support staff (4) | – |
| Facility 4 | – | Support staff (5 members) |
| Facility 5 | Support staff (1) | – |
| Total | 11 | 2 |

[a]Support staff included- Staff nurse, pharmacist, lab technician, health supervisor, multi-purpose health workers.

We also did interviews and discussions at primary health care facilities. We purposively sampled five facilities that had acquired certification in last one year. All facilities were from rural settings. All facilities had the full complement of staff present. The staff at one of the facilities had also served as 'mentors' to other health facilities that were applying for certification. We discussed with staff about their engagement with the NQAS certification process, asking for concrete examples of the issues they faced and overcame during the process. We also asked their opinion on whether the certification made any difference to their work at the health facility thereafter.

The interviews and the FGDs were conducted by researchers from the Indian Institute of Public Health Gandhinagar, supported and trained by John Hopkins University. The interviews lasted 30 to 45 minutes, and FGDs were of 45 minutes-1 hour duration. Data collection was done in both languages, Gujarati and Hindi. Most interviews were recorded. Four participants did not consent for the interviews to be recorded.

As is customary in qualitative research, the precise sample of participants was not predetermined. We stopped recruiting new participants when we achieved data redundancy, that is, no important new themes emerged from ongoing analysis. Finally, we did a member check with one state and one district policymaker to validate our findings.

## Data analysis

The analysis of the data was initiated concurrently with the data collection, as is the practice in qualitative studies [24]. Data debrief sessions to discuss notes from the field were held routinely during data collection. All recorded interviews were transcribed to English before analysis, and the field notes for the non-recorded interviews were typed out. We used generic thematic analysis methods for the analysis [24,25]. Initially, SR open coded four transcripts and derived an initial codebook, based on the transcripts and the interview guide. Following this, SL and PV independently coded the transcripts and notes line-by-line. Discussions were held within the team to resolve discrepancies in coding and refine the codebook. In addition, HJ also coded transcripts at the primary health facilities to deepen the thematic area on the contribution on certifications to quality. The open-source software, Taguette (https://www.taguette.org/) was used to aid the analysis.

We developed preliminary analytical summaries for all themes in the codebook. Our preliminary analytical summaries for the first research question (to draw policy lessons on factors leading to the successful certification) appeared to resonate with the broad themes of the policy triangle framework (Fig 4) [18]. Hence, we used this framework to deepen the analysis in our study. For the second research question-on how certification processes contribute to improving the quality of care at primary health care facilities- we clustered the respondents' perspective into four inductively derived themes- improvements in infrastructure

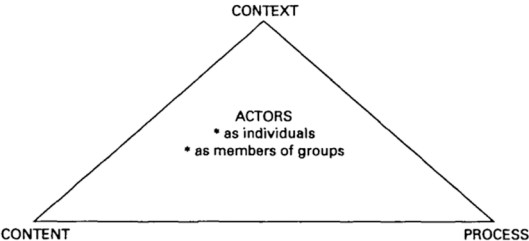

**Fig 4. The health policy analysis triangle [18].**

and ambiance at health facilities, improvements in provider competencies, improvements in organization, and improvements in collaboration with and across facilities.

## The health policy analysis triangle

The policy triangle framework has been widely used to study diverse health topics and is pivoted on the idea that much more than the mere 'content' of policies influences the way they play out in healthcare settings [26]. The triangle comprises four domains: content, context, processes, and actors [18], which interact with one another:

- Actors refer to stakeholders- policy makers, managers, implementors- who have power to influence the way the policy works individually or in collaboration.

- Processes refer to how the policy gets developed, planned, implemented and evaluated, to achieve its vision.

- Content refers to the meat of the policy and how it gets interpreted by actors.

- Context refers to the cornucopia of social, political, cultural, and other environmental factors that influence policies in different ways.

This framework has been used across the policy continuum- from agenda setting to implementation [26].

## Ethical statement

Ethical approval for the study was obtained the institutional ethical review committees at the Indian Institute of Public Health, Gandhinagar, and from Johns Hopkins University (IRB Number- 21610). Verbal consent was taken from all participants before the data collection.

## Findings

In the first section of our findings, we delineate factors that have enabled the successful uptake of NQAS certification strategies in Bhavnagar. We present these factors as interactions between actors and other three prongs of the policy triangle- context (why), content (what), and processes (how). Fig 5 shows an adapted version of the policy triangle to depict the key themes in this study.

In the second section of our findings, we summarise the perspectives of staff on whether certification processes have contributed to improving the quality of care in primary health care facilities in Bhavnagar.

## Section 1: Factors that have enabled the successful uptake of NQAS certification strategies in Bhavnagar

**1A. Support from the district and state leadership (actors-policy context).** We found that the district managers in Bhavnagar strongly endorsed the NQAS certification initiative. The initiative was regarded as being outcome-oriented and fitted well into the usual modus operandi of the Bhavnagar administration, that tended towards being planned, focused, and directive. There was a strong sense of competition with respect to NQAS certifications, and district managers wanted to outdo other districts in the state in obtaining numbers of certificates.

The supportive policy context for NQAS in the district was apparent in many ways in our discussions. First, the district managers tried to ensure the quick disbursement of funds to facilities that have been lined up for certification. Second, it was reported that the managers often visited facilities personally to provide assurance of support in certification endeavors, as

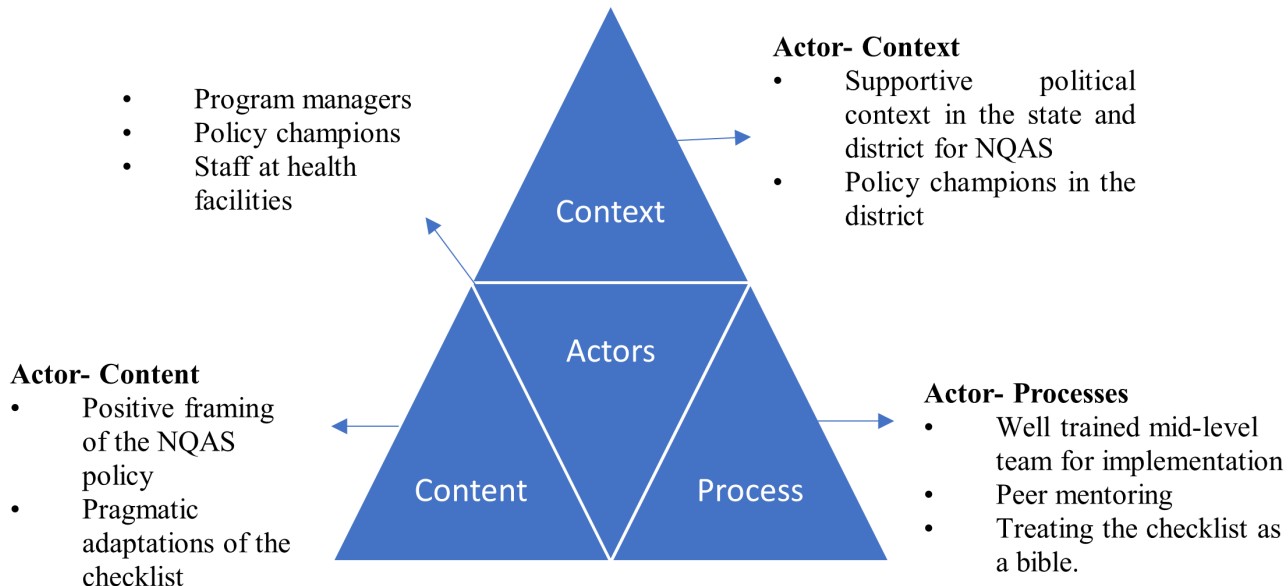

**Fig 5. Analytical framework for the study.**

well as exert soft pressure on the staff. Third, the managers reported both planning for achieving certifications at the district level as well as closely monitoring these plans. Fourth, the senior management has set up a mid-level operation team in Bhavnagar to direct and oversee the NQAS certification process at facilities (see Box 1 for quotes on the supportive policy context). Bhavnagar also has a history of engaging with different quality certification initiatives. The district had engaged early-on with a sister quality certification initiative, Kayakalp, that is focused on hygiene, patient safety and satisfaction in health facilities. Having the precedence of participating in the Kayakalp certification programs was also reported to contribute to a supportive policy environment for NQAS certification in Bhavnagar.

At the state level too, in Gujarat, quality certification strategies were widely supported. This was evident from existence of state-level initiatives like the '*100 Days and 100 NQAS scheme*' (wherein 100 facilities were taken up for certification in 100 days), the provision of training and weekly reviews with district level leadership on quality. The state quality cell has also supported the district through timely disbursement of funds and the issue of directives on NQAS.

> **Box 1. Quotes to illustrate the political context in Bhavnagar that was supportive of NQAS certification.**
>
> *"We never stopped at one-time training of NQAS for the facility staff. There were repeat trainings. Whenever we visited facilities, and if a staff did not know a particular thing, we did on the spot training there itself"* (District program manager, Female)
>
> *"We received support from the district authorities. When we applied for NQAS, I requested district madam that I need someone to handhold me, and I cannot do it alone. She arranged for deputation of staff from nearby NQAS certified facility for two-three days."* (Support staff at facility 3, Male)

> *"The checklists for 5S, Kayakalp, and Kaizen cover many of the same topics as the NQAS checklist. So, if you have already completed one of these certifications, it becomes a good foundation for NQAS." (Doctor at facility 2, Female)*

**1B.  How policy champions engaged with the NQAS policy (actors-policy content).** Within the district administration, Bhavnagar has had strong proponents of the NQAS certification initiative. Two such champions of the NQAS initiative were named repeatedly during our discussions with the staff. These champions had belief in the logic of certification, as well as a high-level of agency and political voice in the district to advocate for the funding needed for NQAS certifications of health facilities. Further, they had technical knowledge about the NQAS checklist, that served as the basis for facility assessments.

Policy champions in Bhavnagar had motivated and trained a set of mid-level operational leaders to undertake and scale up NQAS. Box 2 highlights the perspective of one such champion in Bhavnagar.

## Box 2. A policy champion's perspective on NQAS certification.

*It is important to connect with people emotionally. So, the first thing I would do is to have a rapport with staff (in the facilities)- just visit them, don't talk of NQAS. I used to take milk from Bhavnagar and go to the remote areas, since there was no milk there and tell them- I am here for having tea. I would appreciate the tea. This is my approach. Throughout India, the staff is reluctant to participate in NQAS- but that is because the district level teams are not sensitizing them to the importance of NQAS. If you start riding a car, you have to start in first gear- it will take time to pick up. But when it is in 5th gear, that is when you will start saving petrol. I tell people that we are in first gear now. But we must start there to reach 5th gear. We must make them understand why NQAS is important.*

*I have also maintained all high-level relationships. It was important to make myself indispensable to my senior manager. If the seniormost manager is not able to perform anything, he would call me, and I would never say no to him. So, in return, when I wanted things done, they would never say no. I had gained their trust.*

*I have established a good team here. I have planted the seeds here, now the trees have to grow. I tell my training team that training doesn't mean talking about the checklist alone with the facility staff. The training must be about how to change mentality. Also, step by step, we need to introduce the checklist to staff (in the facilities). If you overfeed someone, he will only vomit.*

**1C.  Well-trained mid-level team for implementation (actors-policy processes).** Bhavnagar currently has a strong mid-level operations team of 5–6 people (mostly doctors from alternative medicine streams) who closely handhold facilities undergoing NQAS certification. Each facility gets assigned to one person from the mid-level operation team. The assigned person pre-assesses the facilities for potential to obtain certification. Once a facility has been selected, the assigned person is deputed there temporarily, and does a variety of tasks: such as dividing the work to be done among staff, identifying improvements needed in facilities for obtaining certifications, finding pragmatic solutions and workarounds to overcome shortcomings in health facilities. The formal training of staff is also done by this team on different aspects of NQAS. In addition, members of this team provided informal

training of diverse nature, including for structural improvements (where to position the fire extinguisher or put up the health promotion material) and for process modifications (like how to mop the floor correctly and prepare documentation meticulously).

The mid-level team also does mock interviews with the facility staff, in preparation for the external assessments. The constant support from the mid-level implementation team was widely appreciated by staff in all our discussions (see Box 3 for quotes).

---

**Box 3. Quotes to illustrate the support from mid-level manager's team in NQAS certification.**

*"Earlier, we were a team of only three people at the district level (for implementing NQAS). Then, we included 5–6 AYUSH MOs who were trained in NQAS. They visit facilities, check and support them, and give me complete feedback in the evening". (District program manager, Female)*

*"Dr. X (mid-level manager) was with us during the NQAS training. So far, whatever difficulty has happened, he tells us what to do. So that more and more work gets done. There are shortcomings in which sir guides when he visits facility" (Focus group discussion at facility 1)*

---

**1D.  The positive framing of NQAS by all actors (actors-policy content).**  One of the biggest contributions of the policy champions, and the trained mid-level operations team is the positive framing of NQAS throughout the district. These actors have inculcated a perception of NQAS certifications as actionable within the administrative structure of the district.

*"NQAS creates positive vibe, a positive environment in the facility. It looks clean. There is team-bonding because people have to work together for a long time to achieve the certification. It leads to team spirit. The provider's knowledge and skills also improve because it is a training. Even attitudes, I have seen change. There is a sense of achievement when the certification is achieved. They get recognition, their photo comes in TV, WhatsApp" (District program manager, Male)*

In most interviews, people have acknowledged the usefulness of NQAS certification process. Staff have shared that participating in these certifications has improved the appearance of health facilities, given staff a boost in confidence and skills, and improved several routine operations (such as labelling of drugs) in facilities.

**1E.  Treating the checklist as a bible (actors-policy processes).**  The district has adopted a pragmatic approach to completing the official checklist for NQAS certification. They focus on getting 70% of the checklist (the minimum required for certification) completed rather than aiming for 100% improvement. Respondents at all levels have highlighted the importance of using the checklist as a 'bible'. One key point that was repeated during our discussions was that the facility staff needed to be trained on the checklist and that the 'fear of the checklist' needed to be dispelled (see Box 4).

---

**Box 4. Quotes to illustrate the importance of following NQAS checklist as perceived by respondents.**

*"We used the NHSRC based application (online checklist) to identify loopholes and team building to ensure proper work distribution". (Staff at facility 5, Male)*

*"Fear of the checklist needs to go from the minds of staff in the facility (Policy entrepreneur interview)*

---

> *"The first thing we do is start training the staff on the checklist. This is the most important, because it helps us to identify gaps in the facilities. Once the gaps are identified, we can work to fill them. (Sub-district program manager, Male)*

**1F. Pragmatic adaptations to meet checklist requirements (actors-policy content).** The political will to obtain certifications has led to several pragmatic adaptations of the NQAS strategy on the ground. Many of the adaptations we saw were spearheaded by the mid-level implementation team and supported from district level managers. Discussions with staff from the mid-level implementation team showed that facilities intended for certification often lacked the necessary infrastructure, human resources and funds. But the implementation team had learnt to salvage infrastructure, share equipment between facilities, ask for donations, facilitate temporary deputation of additional staff, in order to, one way or another, fulfil the requirements of the checklist. The attitude of 'not giving up', endorsed by the policy champions, had trickled all the way down to the facilities, resulting in several make-dos and make-shift arrangements that enabled certifications.

Some examples of pragmatic adaptations done to enable certifications are tabulated in Box 5.

---

### Box 5. Examples of pragmatic adaptations of the checklist.

Adaptations to make up for deficits in infrastructural requirements in facilities for NQAS: *rearranging furniture at pharmacies to improve patient access; setting up a make-shift fire escape putting screens between rooms when constructing additional rooms was not practical, enabling infrastructure sharing between facilities.*

Adaptions pertaining to human resources: *Temporary staff deputation to help with additional work during the certification process.*

Adaptation to improve local understanding of checklists: *Translating documents to local language for easier understanding.*

Local budget adjustments: *integrating training sessions with regular meetings to save on funds; using funds interchangeably between facilities.*

Help from local communities: *asking for donations and local support to fulfil requirements of the checklist (one specific example included asking an ice-cream shop owner to donate a digital board to the facility)*

---

**1G. Peer mentoring to enable knowledge-sharing (actors-policy processes).** Bhavnagar has a peer-mentoring setup for primary health care facilities, wherein staff from an already-certified health facility act as mentors to staff from facilities that have been chosen for certification. As part of the peer-mentoring process, cadre-to-cadre relationships have been built. For instance, a pharmacist from a certified facility mentors a counterpart at the prospective facility. The team that is applying for certification is also taken to other already-certified facilities for learning visits. Staff reported finding the peer-mentoring process very useful:

> *"We were guided by those staff who already had experience of the NQAS process before one or two month of the inspection that how to maintain record and register, how to do labelling and all the things like cleaning and infrastructure". (Staff at facility 2, Female)*

*"In this, the pharmacist from the other health center guided me and helped me a little. Their NAQS had been done. They guided me and helped we arrange the vaccines perfectly". (Staff at facility 3, Male)*

Peer-mentoring facilitated the sharing of knowledge between facilities. It helped to form informal quality circles where several real-world, practical certification-related concerns were discussed. Staff from prospective facilities often dreaded the external review process (wherein they would be interviewed by external assessors); peer-mentoring also provided a way for facility staff from prospective facilities to informally discuss about the external assessment process with those with experience. Further, peer-mentoring also encouraged soft competition between facilities, and instilled a desire among staff in prospective facilities to obtain the certification.

### Section 2: From NQAS certification to improved quality of care

In this section, we share the perspectives of staff in Bhavnagar on whether NQAS certification has led to improvement in the quality of care provided by facilities. Fig 6 summarises these perspectives.

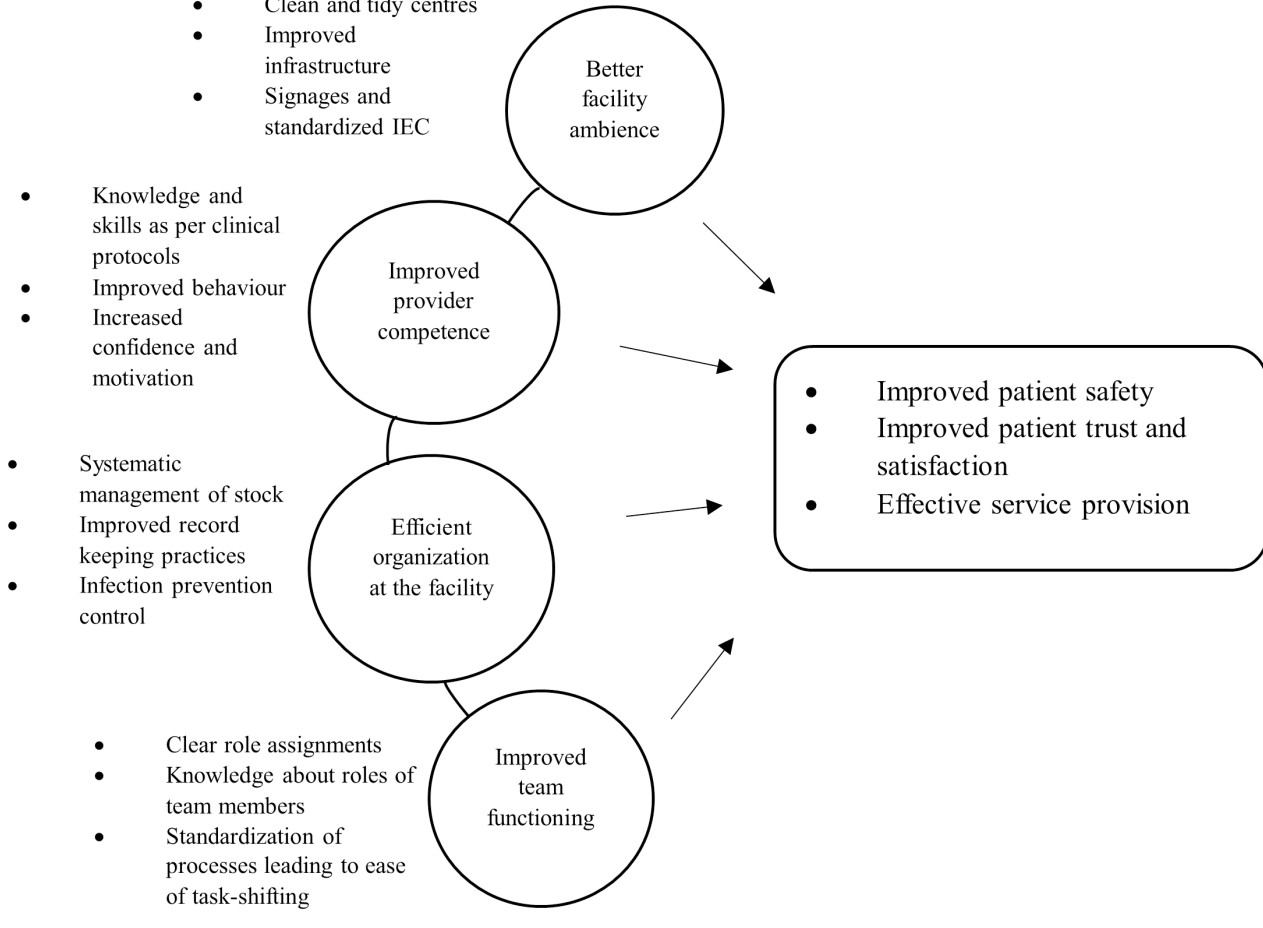

**Fig 6.  Perceptions of respondents on the usefulness of NQAS certification.**

**2A. Perspectives on the uses of NQAS.** We have synthesized the perspectives shared by staff on the uses of NQAS under four themes:

*Improved facility infrastructure and ambiance*: Staff shared that the process of doing certifications catalysed action on infrastructure improvements such as furniture upgrades, electrical repairs, painting, and pest control measures. Communication material put up in the facility was thought of as appealing to the community. Overall, it was felt that the implementation of NQAS improved the appearance of healthcare facilities, making the facility a more appealing workplace for staff, as well as enhancing patients experience of care.

> "*This is our workplace- we spent majority of our time here (at facility), so this place affects us lot, and (clean and organized) facility gives us motivation*" (Doctor at facility 1, Female)

> "*Earlier the people did not know which programs are being run by the government, but now because of IEC, they know which programs are currently implemented at the facility and what benefits they can get*". (Focus group discussion with staff at facility 1)

*Improved provider competence*: Staff reported a boost in confidence and skills following NQAS training, which subsequently upgraded their ability to provide effective patient care.

> "*If a patient comes with an infection, we will now follow protocols. We will not get infected, and their treatment will also be good. And the safety of both will be maintained*" (Doctor at facility 3, Female)

> "*After NQAS, I am managing drug stocks efficiently, ensuring minimum stock levels, and proactive management to prevent stock outs.*" (Staff from facility 2, Male)

*Efficient organization*: Staff shared participating in NQAS led to refinements in infection control protocols, hygiene practices, and overall cleanliness, promoting a safer environment for both patients and staff. Furthermore, NQAS implementation was perceived to improve various operational aspects of facilities, including stock maintenance, medication management, record-keeping, documentation, and the presentation of work activities. The checklist-based approach of NQAS process was reported to introduce standardization in inventory management, reducing errors and enabling communication between staff.

> "*Earlier if there was bleeding, we used to just wipe it, but now we follow proper protocol in case of blood spillage. We interact with the patient while also maintaining safety.*" (Focus group discussion with staff at facility 1)

> "*We have become more systematic, like placing medicines, documents, and narcotic drugs in designated locations*". (Staff at facility 2, Male)

*Improved collaboration within and across facilities*: NQAS implementation encouraged team collaboration among the different levels of healthcare, resulting in increased cross-referrals and improved follow-up procedures due to better record-keeping practices. It also facilitated departmental organization, clarified staff duties, and standardized processes. Policy-level actors also cited enhancement in safety measures and fire evacuation training, ensuring a safer environment for staff in health facilities:

> "*Teamwork is mandatory for achieving NQAS certificate. We have to discuss, initiate things and share work department-wise*" (Focus group discussion with staff at facility 4)

*"The advantages (of NQAS certification) include improved team bonding and support from the villagers". (Staff at facility 5, Male)*

Overall, certification was perceived as being useful to staff in terms of personal growth, having a better workplace and enhancement in teamwork. All this, it was shared, was likely to lead to better service provision on the facility, and improvements in patient safety. A few respondents also shared that the utilization of services by patients improved after NQAS, since the ambiance at the facility was more welcoming to patients; and the clean, newly painted buildings signalled to the community that the facility was likely to be as 'good' as a private one.

*"Patients also like NQAS because the health centre now seems like a private hospital to them"* *(Doctor at facility 3, Female)*

*"Since NQAS (certificate is received) there are more patients coming and institutional deliveries are also increased" (Support staff at facility 2, Female)*

**2B.  Challenges faced in improving quality through NQAS certification.**  As mentioned earlier, the NQAS checklist was treated as a 'bible' for obtaining certifications. While this approach enabled quick certifications, the broader importance of NQAS as a means to improve quality of care sometimes got de-emphasized in the district due to an intensive focus on completing the assessments. Policy champions often tried to delicately balance the need for being in 'examination mode' to complete the certification with the need to foster a culture of quality in facilities.

Some other limitations of the NQAS certification process in improving quality of care was also mentioned by staff at facilities. It was shared that this particular certification initiative focussed mainly on improving facilities and emphasized less on improving processes of engagement with communities, which was also key to improving overall quality. Further, the certification process was regarded as time-consuming, involving a temporary increase in the workload of staff. In places where key staff positions (medical officers, nurses, pharmacists) were vacant, the existing staff felt overloaded during the certification process.

*"While working with patients, if any other urgent work comes up related to NQAS, then you have to do that. So, it is a bit disturbing, and that's the reason we need additional staff"* *(Focus group discussion with staff at facility 4)*

*"There is (as such) no problem in doing NQAS. But in the preparation for all of these, the daily routine programs that facilities have to do get disturbed". (Sub-district program manager, Male)*

Staff reported challenges in maintaining the processes after certificate has been achieved and mentioned the need for additional resources for sustaining the certification.

*"We will need additional staff (staff nurse) as our workload is increasing after NQAS. Refresher trainings in updated protocols and more funds will be useful to maintain the practices" (Focus group discussion with staff at facility 1)*

It was also pointed out that while quality certifications could improve procedures and protocols, and contribute to patient safety, these initiatives could not solve important foundational challenges in the health system- such as persistent human resource deficits, target-oriented mentalities that led to siloed work, and drug stock outs. Persistent

foundational challenges in the health system remained even after certification. These challenges undermined the sustainability of the advantages brought about through quality certification initiatives.

## Discussion

Quality improvement has become an integral part of policy agendas of national health systems across the globe [27,28]. However, studies that engage deeply with the implementation of quality improvement strategies are lacking [29]. This empirical case-study from Bhavnagar district in Gujarat aims to understand factors contributing to the successful quality certifications is an attempt to fill this gap. In Box 6, we have consolidated key learnings from this study, that could be useful in boosting certification processes in similar geographies.

---

**Box 6. Learnings from Bhavnagar on implementing quality certification initiatives in districts.**

**High level management must plan for and direct certifications**

- High level managers in the district should ensure the timely release of funds to facilities.
- They should establish a mid-level operation team to guide the certification process.
- They must identify policy champions and empower these champions to work with the mid-level implementation teams.
- They should plan and monitor the process.

**Policy champions must be identified within a district**

- Champions who believe in quality and can be given protected time to take the certification process forward must be identified.
- Champions must create a positive image of certification processes, motivate operational teams, and contribute to strategic planning.
- Champions should be well-versed with the checklists and assessment processes.

**A robust mid-level implementation team is needed**

- A team of mid-senior-level people who can handhold the facilities closely must be put in place.
- This team should provide on-the-ground support to facility staff during the certification process, supervise and motivate facility staff in preparation for certification. This team should also provide day to day troubleshooting support (both online through WhatsApp groups and other means, and via face-to-face visits). The team should act as a link between the facility and the district.

**Peer mentoring mechanisms can enable learning across facilities and motivate staff.**

- Motivated staff from already-certified primary health centers should be identified as staff mentors.
- Staff mentors must visit facilities that have been lined up for certification to handhold them.
- The team applying for certification can also visit other already-certified facilities for cross-learning.

---

> **Pragmatic adaptations of implementation processes must be done**
>
> - Each facility must have at least one person from the mid-level operation team who pre-assesses facilities, divides work, identifies gaps, and enables the facility to pragmatically work with certification checklists.
>
> - The checklists must be pragmatically adapted so that facilities can best use the strengths they have to work towards obtaining the certification.
>
> - Training must focus on removing the fear of checklists.

The lessons consolidated in Box 6 illustrate the usefulness of a 'positive deviance' approach [30] to unearthing factors that underlie quality initiatives at primary health care level in health systems. The use of this approach has enabled nuanced understandings of what worked in this setting, that can be replicated elsewhere, and thereby bring improvements at scale [15].

Our findings emphasize the importance of having a proactive state and district environment to advance the uptake of quality improvement interventions. The need to 'govern' for quality [31,32] and have the political leadership pay attention to quality, have been emphasized repeatedly in literature [33]. The demand for quality in health systems does not automatically occur but needs strong advocates at all levels [16]. Despite this recognition empirical documentation of the political barriers and facilitators of quality improvement are currently lacking. In our study, the proactive policy environment in Bhavnagar, with both state and district authorities actively promoting NQAS certification, led to the prompt allocation of funds, structured planning, and delegation of tasks to a robust team to support operations. We also found strong champions in the government for the NQAS policy. These champions played a role akin to 'policy entrepreneurs' [23], having the agency, persistence, and the technical know-how to enable policy success.

We also encountered many pragmatic ways in which the NQAS process was adapted to the needs of each prospective facility selected for certification. These adaptations included adjustments made to funding, the deployment of additional human resources as needed, and ensuring the best use of existing infrastructure. The "*never give up*" attitudes of the mid-level operations teams had permeated the facilities, leading to several improvisations and workarounds that made certifications possible. These improvisations were typical examples of 'street-level' discretion, and its influence on policy. It is widely acknowledged that discretion at street-level can mediate both the success and failure of policy and modify local leadership and motivation [34]. Studies from other contexts too have emphasized on the importance of local motivation, positive attitudes towards problem solving among staff, and the ability to manage relationships for quality improvement in healthcare facilities [35].

Peer mentoring of facilities emerged as a powerful mechanism for knowledge exchange and experiential learning in our study. There is less literature on facility-based, peer mentoring for quality improvement in similar contexts. We only found one other study from Cambodia where the quality improvement strategy involved explicit peer-to-peer learning to improve performance [36].

The impact of quality certification initiatives on quality of care has been a topic of important debate in the recent years. While accreditation and certification strategies have been widely used across countries, there have been examples such as SafeCare in Tanzania [37], and Manyata in India [38] that suggest that these strategies make limited contributions to improving quality outcomes. In India, local studies from Karnataka, Maharashtra and Chhattisgarh have indicated improvements in some aspects of quality at health facilities- workflow, medical care, and an increase in the number of patients accessing outpatient care- after NQAS

certification [39]. These findings align with the perceived advantages of certifications stated by respondents in our study as well. However, the recently launched certification processes in India are yet to be evaluated at scale.

Our discussions in Bhavnagar suggest that the intensive focus on the checklist, while aiding quick certification, can become reductionist in its approach to improving quality in health systems. The district management team and policy champions attempted to inculcate a culture of quality in the health facilities using the tool of NQAS certification. This is observed to some extent in the perceived understanding of the staff of the term 'quality of care', that includes improved patient safety and satisfaction and better health care. However, in the due course, the certification process itself took precedence over the efforts for bringing any deeper meaningful changes driven by values of providing good quality care as people's right. Other studies have pointed out this challenge as well, that the value of external review processes for similar initiatives can get reduced to being a "*summative evaluation rather than a mechanism for formative learning*" [4]. Despite this, our field discussions suggest that the process of quality certifications can enable learning and teamwork, improve infrastructure and ambience in facilities and improve protocols and measures for patient safety in facilities. Some of these improvements, particularly those that don't find meaning in staff routines, may however, only be temporary.

Moving forward, there is a need to balance the tangible elements of the certification process (obtaining the certificate and the incentives attached) with the intangible ones (such as providing an environment for continuous learning and problem-solving through the certification process). The worth of NQAS must be perceived as more than merely obtaining a certificate. The extension of the current NQAS policy to provide post-certification support could be useful for enhancing the sustainability of efforts. Finally, there is also a need to address foundational gaps in quality in health systems- attrition in human resources, infrastructure deficits and competing demands from a variety of programs in health facilities. These gaps are found across health systems in resource-constrained settings [40–42]. It has been noted that foundational structural reforms in health systems and building linkages with other social systems is critical to improving quality [43]. Thus, working on NQAS certifications can be one of the routes to strengthening the foundation of good quality health systems in the country, but it cannot be the only route.

This study has some limitations. It is a single case study from Bhavnagar, and we do not have comparators to validate our findings against. Further, we mainly relied on people's memories of the NQAS certification process, and there could have been bias in recall. We did not collect data to verify the benefits of NQAS reported by the staff on improved service utilization or facility management. As the study focused on perceptions of stakeholders on the NQAS process, we did not include questions on compliance beyond the NQAS audit. Since this study was designed as a 'positive deviance' study, we have emphasized less on the limitations of NQAS (though we did add some points in section 2 of the findings). Despite these limitations, this study is context- sensitive and captures rich experiences with quality certifications from the public health system.

Finally, the use of the policy triangle [18] as an overarching analytical framework in this study also demonstrates it evergreen usefulness [18]. Our findings illustrate how the different components of the triangle- actors (program managers, policy champions and facility staff engaged at various steps and influencing the processes), processes (planning and implementation of the NQAS policy), content (NQAS guidelines and checklist) and context (supportive environment at the district)- are intertwined with one another to influence policy success. This study underscores that success of the NQAS certification initiative in Bhavnagar is likely to elude simplistic explanations, and a whole array of intersecting factors are responsible for the district's excellent performance on NQAS.

## Conclusion

This study delineates various factors that have led to the successful quality certifications of health facilities in one district in India- Bhavnagar. The use of the policy triangle as an overarching analytical framework helped unearth the enabling factors that lie beyond the content of NQAS policy itself, and that have broader implications for other districts and states in India. Out study highlights the following factors as key to the successful implementation of quality certifications. First, a strong district level management team is required for planning, funding and monitoring the certification process. Second, policy champions who believe in quality and have in-depth knowledge of the process need to be identified. Third, a robust mid-level implementation team is required for handholding the facilities closely and providing day-to-day support through various communication channels. Forth, experiential learning is likely to be more helpful than top-down approaches of training for quality. Peer mentoring mechanisms (from already certified facilities) enable such learning across facilities and staff and help in improving motivation. Fifth, pragmatic adaptations of the implementation process that build on strengths of local health systems and suit the contextual requirements of the facility/ district are required. This implementation case study thus provides insights that can help in the successful uptake of the quality certification policies in real-world contexts.

## Supporting information

**S1 File. Tools used for data collection.**
(DOCX)

## Acknowledgments

The authors would like to thank the participants in this study for contributing their time and their experiences to this research. We are grateful to the Bhavnagar district health management team, particularly Dr. Manasvini Malaviya and Dr. Dhaval Dave for facilitating the study proceedings. We acknowledge Dr. Deependra Dube and Dr. Pratiksha Ganasva from Indian Institute of Public Health, Gandhinagar for help with data collection.

## Author contributions

**Conceptualization:** K. Shruti Lekha, Sudha Ramani, Harsha Joshi, Chandramani Kumar, Ankita Shah.

**Formal analysis:** K. Shruti Lekha, Sudha Ramani, Preet Verma.

**Investigation:** K. Shruti Lekha, Sudha Ramani, Harsha Joshi.

**Resources:** Chandramani Kumar.

**Supervision:** Sumit Malhotra, Tapasvi Puwar, Chandramani Kumar, Anish Sinha, Deepak Saxena.

**Writing – original draft:** Sudha Ramani, Harsha Joshi, Ankita Shah.

**Writing – review & editing:** Sumit Malhotra, Tapasvi Puwar, Anish Sinha, Deepak Saxena.

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
