## [Decision Letter · Decision Letter 0]

22 Oct 2024

PGPH-D-24-02015

What enabled the successful implementation of a quality certification initiative in Bhavnagar, Gujarat? A policy analysis case study

Dear Dr. Harsha Joshi 

Thank you for submitting your manuscript to PLOS Global Public Health. After careful consideration, we feel that it has merit but does not fully meet PLOS Global Public Health’s publication criteria as it currently stands. Therefore, we invite you to submit a revised version of the manuscript that addresses the points raised during the review process.

We look forward to receiving your revised manuscript.

Kind regards,

Shweta Marathe, Health System Researcher

Guest Editor

Journal Requirements:

1. We would like to request copy editing.

Additional Editor Comments (if provided):

- This is an important empirical study, and the paper is overall well-written.

- While the positive model provides valuable insights, it’s equally important to analyse failed models to understand what didn’t work. Authors may explain rationale in studying only positive model.

- Please clarify the sampling method used for selecting the five PHFs and specify whether they were from urban, rural, or mixed settings.

- In Table 3B, facilities 1-5 are mentioned, but later it states that four PHFs were selected. Please correct or clarify this inconsistency.

- In line 195 of the analysis section, there is some repetition regarding the frameworks mentioned. Kindly eliminate the repetition references.

- The initial part of the discussion (lines 440-452) would be more appropriate in the introduction, particularly in the literature review or rationale section. The authors might consider relocating this content.

- In line 528, please clarify the different components of the triangle—actors, processes, and context—while removing the repeated word "context."

- It would be beneficial to shorten the discussion section and include a concise conclusion to enhance readability of the manuscript.

Reviewers' comments:

Reviewer's Responses to Questions

**Comments to the Author**

1. Does this manuscript meet PLOS Global Public Health’s publication criteria ? Is the manuscript technically sound, and do the data support the conclusions? The manuscript must describe methodologically and ethically rigorous research with conclusions that are appropriately drawn based on the data presented.

Reviewer #1: Yes

Reviewer #2: Yes

2. Has the statistical analysis been performed appropriately and rigorously?

Reviewer #1: Yes

Reviewer #2: Yes

3. Have the authors made all data underlying the findings in their manuscript fully available (please refer to the Data Availability Statement at the start of the manuscript PDF file)?

Reviewer #1: Yes

Reviewer #2: Yes

4. Is the manuscript presented in an intelligible fashion and written in standard English?

Reviewer #1: Yes

Reviewer #2: Yes

5. Review Comments to the Author

Reviewer #1: Discussion Section:

The discussion effectively draws from the findings, but you may want to emphasize the broader implications for other districts or regions in India. A few more explicit recommendations could be added in this section to strengthen the conclusions.

Since the study uses the "policy triangle framework," you might want to summarize this framework earlier in the introduction to familiarize readers with it before the results section.

Conclusion:

The conclusion is clear but can be enhanced by offering more concrete suggestions for scaling the findings across other districts in India or similar health settings globally.

Reviewer #2: Congratulations on being able to conduct an important study.

Here are my thoughts and queries:

“In this study, we have used the policy triangle to deepen our understanding of the factors that influenced the implementation of NQAS in Bhavnagar” – need better justification of why the policy triangle framework was used. Why not a framework like “theory of change” that suits the context better?

What is done if some parameters remain unmet, and how is compliance ensured/maintained beyond the audit? It should have been included in questions/ “topics” covered in data collection to both district-level actors and facility staff. What is being done to ensure a sustained change (in terms of processes, behaviour, competence) in the way of providing care (clearly hiring ad hoc staff, or getting donations from some community members isn’t sustainable)?

One of the PHC staff said that they have been able to improve inventory management, prevent stockouts etc. How exactly was the staff member able to do that? How did a checklist based approach helped the staff member in improve the process?

“Throughout India, the staff is reluctant to participate in NQAS- but that is because the district level teams are not sensitizing them to the importance of NQAS.” What approach did the Bhavnagar team took in sensitizing the staff?

Study shows that a sense of achievement, as well some healthy competition (for monetary and non-monetary reasons) motivates the staff. But is that enough? Authors could have also asked if there is any deeper meaning that drives them to develop policy framing of NQAS. I would be inclined to think the understanding that “quality matters because people’s well being, and rights matter” could really drive meaningful change by inculcating these values that could be imparted to health providers (beyond a certification process)

“the facility staff needed to be trained on the checklist and that the ‘fear of the checklist’ needed to be dispelled” how was it identified that there is a fear of the checklist? Any assessment was done to understand staff apprehensions/perceptions?

Lastly, did any of the policy actors not see a need for better community engagement that just settling for said community satisfaction?

6. PLOS authors have the option to publish the peer review history of their article (what does this mean? ). If published, this will include your full peer review and any attached files.

**Do you want your identity to be public for this peer review?** For information about this choice, including consent withdrawal, please see our Privacy Policy .

While revising your submission, please upload your figure files to the Preflight Analysis and Conversion Engine (PACE) digital diagnostic tool, https://pacev2.apexcovantage.com/ . PACE helps ensure that figures meet PLOS requirements. To use PACE, you must first register as a user. Registration is free. Then, login and navigate to the UPLOAD tab, where you will find detailed instructions on how to use the tool. If you encounter any issues or have any questions when using PACE, please email PLOS at figures@plos.org. Please note that Supporting Information files do not need this step.

---

## [Editor Report · Decision Letter 1]

9 Dec 2024

PGPH-D-24-02015R1

What enabled the successful implementation of a quality certification initiative in Bhavnagar, Gujarat? A policy analysis case study

Dear Dr. Joshi,

Thank you for submitting your manuscript to PLOS Global Public Health. After careful consideration, we feel that it has merit but does not fully meet PLOS Global Public Health’s publication criteria as it currently stands. Therefore, we invite you to submit a revised version of the manuscript that addresses the points raised during the review process.

We look forward to receiving your revised manuscript.

Kind regards,

Johanna Pruller

Staff Editor

PLOS GPH

on behalf of 

Shweta Marathe, Health System Researcher

Guest Editor

Journal Requirements:

Additional Editor Comments (if provided):

Kindly make below mentioned minor corrections-

line 172- check and correct table 3 B title

line 189- remove full form of FGD. You have already given it.

Reviewers' comments:

While revising your submission, please upload your figure files to the Preflight Analysis and Conversion Engine (PACE) digital diagnostic tool, https://pacev2.apexcovantage.com/ . PACE helps ensure that figures meet PLOS requirements. To use PACE, you must first register as a user. Registration is free. Then, login and navigate to the UPLOAD tab, where you will find detailed instructions on how to use the tool. If you encounter any issues or have any questions when using PACE, please email PLOS at figures@plos.org. Please note that Supporting Information files do not need this step.

---

## [Editor Report · Decision Letter 2]

30 Dec 2024

What enabled the successful implementation of a quality certification initiative in Bhavnagar, Gujarat? A policy analysis case study

PGPH-D-24-02015R2

Dear Ms Joshi,

We are pleased to inform you that your manuscript 'What enabled the successful implementation of a quality certification initiative in Bhavnagar, Gujarat? A policy analysis case study' has been provisionally accepted for publication in PLOS Global Public Health.

Best regards,

Shweta Marathe, Health System Researcher

Guest Editor